# Inversion of Gravity Data with Multiplicative Regularization Using an Improved Adaptive Differential Evolution

Lianzheng Cheng [1,2,3] , Tiaojie Xiao [4], Xing Hu [5,\*], Ali Wagdy Mohamed [6,7] , Yun Liu [3] and Wei Du [3,\*]

1. Key Laboratory of Critical Minerals Metallogeny in Universities of Yunnan Province, School of Earth Sciences, Yunnan University, Kunming 650500, China; chenglianzh10@mails.ucas.ac.cn
2. School of Mathematics and Statistics, Yunnan University, Kunming 650500, China
3. School of Earth Sciences, Yunnan University, Kunming 650500, China; yunliu@ynu.edu.cn
4. Science and Technology on Parallel and Distributed Processing Laboratory, National University of Defense Technology, Changsha 410073, China; xiaotiaojie16@mails.ucas.ac.cn
5. School of Information Science and Technology, Yunnan Normal University, Kunming 650500, China
6. Operations Research Department, Faculty of Graduate Studies for Statistical Research, Cairo University, Giza 12613, Egypt; aliwagdy@gmail.com
7. Applied Science Research Center, Applied Science Private University, Amman 11937, Jordan
\* Correspondence: xinghu0501@126.com (X.H.); duwei@ynu.edu.cn (W.D.)

**Abstract:** Differential evolution (DE) is a stochastic optimization technique that imitates the evolution process in nature. This paper uses an improved adaptive differential evolution to solve gravity inversion with multiplicative regularization. Compared with additive regularization, the advantage of multiplicative regularization is that it does not require the regularization parameter in the search process. The contributions in this paper mainly focus on two aspects: accelerating the convergence speed of adaptive DE and balancing the effect of model and data misfits in the objective function. The effectiveness of the proposed inversion method is verified by synthetic and field cases. For the synthetic cases, it is concluded that, based on the obtained results and analysis, the presented DE method is superior and competitive with its original version. Additionally, the designed parameter adaptation for multiplicative regularization is useful for trading off the effect of data and model misfits. For the field cases, two successful applications from China were conducted, and the obtained density source distributions were in accordance with those obtained from drilling wells. The synthetic and practical examples demonstrate that high-quality inversion results can be obtained using improved adaptive differential evolution and multiplicative regularization.

**Keywords:** differential evolution; gravity inversion; multiplicative regularization; stochastic optimization

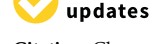



## 1. Introduction

The gravity method is a simple yet effective geophysics technology that has been widely used in various fields such as discovering mineral deposits, petroleum, and geothermal resources [1,2]. In our view, the inversion method of gravity data can be divided up into two types: parametric inversion and physical property inversion. Generally, parametric inversion is used to determine the geometric parameters of a simple source including depth, position, shape, etc. [3–5], while the purpose of physical property inversion is to simulate the density distribution without relying on manual interference [6–9]. As for the physical property inversion, in most cases, the number of optimized parameters is more than the number of observed data. It implies that there exist an infinite number of solutions that can fit the observed data well. So, to reduce the multiplicity, model misfit (also called regularization term) must be introduced [10–12]. At the moment, the inversion objective function comprised of data and model misfit can be solved by local optimization and global optimization. As we know, the local optimization methods (i.e., the conjugate gradient

method [13], Newton's method [14], etc.) have been widely employed. For example, Qin et al. [15] used the non-linear conjugate gradient (NLCG) method to reconstruct the density image of the anomalous body. Using the same method, Feng et al. [16] estimated the basement relief of a rift basin with gravity data. However, the local optimization is easy to fall into the local minima due to its dependence on the initial guess. In contrast, the global optimization starts the search process without requiring good initialization, and can stably converge to a reliable solution [17]. For example, in [17], a particle swarm optimization (PSO) was applied for inverting the basement of the basin; in [18], a modified adaptive differential evolution (DE) was developed for calculating the density of gravity anomaly under the constraint of Lp-norm with $p \in [1, +\infty)$; and, in [19], an efficient genetic algorithm (GA) was developed by Montesinos et al. for efficiently estimating the distribution of density. Apart from PSO, DE, and GA, many global methods are suitable for solving the gravity inversion problem, for example, very fast simulated annealing (VFSA) [20], ant colony optimization (ACO) [21], bat algorithm (BA) [22], etc. From these studies, it can be clearly found that global methods are mainly used for estimating the geometric parameters of field sources. The possible reason is that the physical property inversion is time-consuming. However, compared with geometric inversion, the physical property inversion is more promising since it can recover the shapes of complex sources and depths [23].

Differential evolution (DE) is a population-based meta-heuristic algorithm presented by Storn and Price [24] for solving Chebyshev polynomial problems. Similar to the genetic algorithm (GA) [25], the individuals in DE are updated in terms of Darwin's theory. In recent decades, due to its evident advantages (i.e., simplicity, robustness, speed, etc.), it has been extended to solve optimization problems in many scientific and engineering fields (see [26–28] for more details). In recent years, DE variants have been introduced to invert the geophysical data [18,29–31]. Particularly, the work of [31] indicated that, compared with PSO, DE can achieve slightly better solutions based on robustness, computation cost, and convergence speed. Anyway, as stated by [31], the DE algorithm has not attracted wide attention in the field of geophysical inversion. In addition, these research works indicate that there are few state-of-the-art DE algorithms to be applied in solving physical property inversion. In DE's population, each individual is named as a target vector. Generally, at the first generation, all target vectors of DE are created randomly within the predefined boundary constraints. Then, these vectors are updated through repeatedly conducting mutation, crossover, and selection operations. Usually, mutation and crossover play a vital role in improving the optimization performance. Moreover, the intrinsic parameters including population size ($NP$), crossover rate ($CR$), and scale factor ($F$) also have significant influence on DE's performance since these parameters are capable of balancing the population diversity and the search ability between exploration and exploitation [32]. For the above reasons, a robust and excellent DE algorithm named JADE was designed by Zhang and Sanderson in [33]. Currently, a large number of DE variants inspired by JADE have beaten other evolution algorithms in CEC competitions of single-objective optimization [34]. Hence, this paper will improve JADE for solving the objective function of gravity inversion.

According to the description in [23,35], the data misfit and model misfit in the objective function of gravity inversion are combined together by a regularization factor. So, a proper choice of the regularization factor plays a key role in obtaining a satisfactory solution. Currently, the L-curve method [36], Morozov's discrepancy principle [37], the Unbiased Predictive Risk Estimator [38], and the Bayesian estimator [39] have been developed to estimate the value of the regularization factor. From the literature [36–39], it can be found that all these methods consume intensive computations [40]. In order to overcome the above-mentioned problem, multiplicative regularization was proposed in [41] for the inversion of the contrast source. Later, this method is extended for the inversion of geophysical data in [42–44]. Recently, a general form of multiplicative regularization was proposed by Aucejo and Smet for estimating mechanical loads [45]. The advantage of

multiplicative regularization is that it does not require the regularization parameter in the search process. In this work, the general form of multiplicative regularization is introduced to invert the density distribution of gravity data.

Based on the above reviews and discussions, our main focuses in this paper are twofold: Designing a novel adaptive DE algorithm and inverting gravity data with general multiplicative regularization. For the DE algorithm, the JADE is selected here since it has been proven efficient in solving inverse problems [33]. Considering that the convergence speed is important compared with the global exploration, a new mutation strategy is presented according to the rank value of each vector. In addition, the *CR* value in JADE is randomly calculated under the control of Normal distribution, which ignores the fact that the better information contained in the better individuals is useful for generating superior offspring. Inspired by this observation, a novel *CR* generation scheme in terms of the objective value of each individual is proposed. Furthermore, as discussed in [45], the authors introduced an extra parameter to balance the effect of data misfit and model misfit by using a fixed-point iteration. However, the iteration method is unsuitable for the global search algorithm. Thus, an adaptive selection for the extra parameter is presented. In summary, according to the above discussions, there are three main contributions in this paper: The first direction is accelerating the convergence speed by proposing a novel mutation strategy with rank information. Another direction of this article is introducing a novel *CR* scheme based on objective values in the current population, which can retain the better information of individuals with high rank. Our third direction is developing an adaptive manner to adjust the extra parameter of multiplicative regularization.

## 2. Theory and Method

### 2.1. Enhancement of Differential Evolution

#### 2.1.1. Classic Differential Evolution

In DE, an optimization problem is usually denoted as $\Phi(\boldsymbol{m})$ where $\mathbf{m}$ is a vector with M entries such as $\mathbf{m} = (m_1, m_2, \cdots, m_M)$. For a DE population $\mathbf{P}$, it contains $NP$ target vectors with size $M$. In the population matrix $\mathbf{P}$, a target vector with index $i$ is denoted as $\mathbf{m}_i = (m_{i,1}, m_{i,2}, \cdots, m_{i,M})$. As soon as the initialization is fulfilled, the vectors in $\mathbf{P}$ are updated by repeatedly conducting the DE operations including mutation, crossover, and selection.

- Initialization:

Similar to other meta-heuristic algorithms (MHs), the DE randomly initializes the target vectors within the predefined boundary constraints. For example, if the upper boundary and lower boundary are represented as $\mathbf{m}_u = (m_{u,1}, m_{u,2}, \cdots, m_{u,M})$ and $\mathbf{m}_l = (m_{l,1}, m_{l,2}, \cdots, m_{l,M})$, respectively, then the $j$th entry of $i$th vector is obtained by the following:

$$m_{i,j}^1 = m_{l,j} + rand(0,1) \cdot \left( m_{u,j} - m_{l,j} \right), \tag{1}$$

where $rand(0, 1)$ represents a random number uniformly distributed in the range of $(0, 1)$.

- Mutation:

As a critical operation of DE, the mutation operation is responsible for the generation of the mutant vector. There are several commonly used mutation strategies in [27,46], which can be summarized as the following:

$$
\begin{aligned}
\text{DE/rand/1}: \ & \mathbf{v}_i^G = \mathbf{m}_{r_1}^G + F \cdot \left( \mathbf{m}_{r_2}^G - \mathbf{m}_{r_3}^G \right) \\
\text{DE/best/1}: \ & \mathbf{v}_i^G = \mathbf{m}_{best}^G + F \cdot \left( \mathbf{m}_{r_1}^G - \mathbf{m}_{r_2}^G \right) \\
\text{DE/rand/2}: \ & \mathbf{v}_i^G = \mathbf{m}_{r1}^G + F \cdot \left( \mathbf{m}_{r_2}^G - \mathbf{m}_{r_3}^G + \mathbf{m}_{r_4}^G - \mathbf{m}_{r_5}^G \right) \\
\text{DE/best/2}: \ & \mathbf{v}_i^G = \mathbf{m}_{best}^G + F \cdot \left( \mathbf{m}_{r_1}^G - \mathbf{m}_{r_2}^G + \mathbf{m}_{r_3}^G - \mathbf{m}_{r_4}^G \right) \\
\text{DE/current-to-rand/1}: \ & \mathbf{v}_i^G = \mathbf{m}_i^G + F \cdot \left( \mathbf{m}_{r_1}^G - \mathbf{m}_i^G + \mathbf{m}_{r_2}^G - \mathbf{m}_{r_3}^G \right) \\
\text{DE/current-to-best/1}: \ & \mathbf{v}_i^G = \mathbf{m}_i^G + F \cdot \left( \mathbf{m}_{best}^G - \mathbf{m}_i^G + \mathbf{m}_{r_1}^G - \mathbf{m}_{r_2}^G \right)
\end{aligned} \tag{2}
$$

where subindices $r_1$, $r_2$, $r_3$, $r_4$, and $r_5$ are random integers different from each other within the range of $[1, NP]$, and each of them is not equal to the subindex of $\mathbf{m}_i$. $\mathbf{m}_{best}^G$ represents the optimal vector according to the fitness value at the $G$th generation.

- Crossover:

For a given vector $\mathbf{m}_i$, its trial vector $\mathbf{u}_i$ is obtained by using the following crossover operation:

$$u_{ij}^G = \begin{cases} v_{ij}^G, \text{if } rand(0,1) \leq CR \text{ or } j = jrand \\ m_{ij}^G, \text{otherwise} \end{cases}, \tag{3}$$

where $jrand$ is an integer sampled uniformly from 1 to $M$.

- Selection:

The selection operation of DE is a greedy selection strategy. Generally, the selection operation can be written as:

$$\mathbf{m}_i^{G+1} = \begin{cases} \mathbf{u}_i^G, \text{if } \Phi(\mathbf{u}_i^G) \leq \Phi(\mathbf{m}_i^G) \\ \mathbf{m}_i^G, \text{otherwise} \end{cases}, \tag{4}$$

### 2.1.2. JADE Algorithm

- DE/current-to-$p$best/1 mutation strategy:

In Equation (2), several mutation strategies are listed. Among these mutation strategies, the DE/rand/1 prefers global search over local exploitation, and DE/best/1 is the local exploitive. Inspired by this observation, Zhang and Sanderson proposed the mutation strategy named "DE/current-to-$p$best/1" [33]. The DE/current-to-$p$best/1 mutation strategy can be expressed as:

$$\mathbf{v}_i^G = \mathbf{m}_i^G + F_i \cdot \left( \mathbf{m}_{pbest}^G - \mathbf{m}_i^G + \mathbf{m}_{r_1}^G - \mathbf{m}_{r_2}^G \right), \tag{5}$$

where $\mathbf{m}_{pbest}^G$ is selected from the $p$best set formed by the top $100p\%$ vectors, where $F_i$ is a scale factor associated with the target vector $\mathbf{m}_i^G$.

In order to maintain the population diversity, an archive operation is introduced into the JADE algorithm. Then, another mutation strategy named "DE/current-to-$p$best/1 with archive" was designed in [33], and can be described as:

$$\mathbf{v}_i^G = \mathbf{m}_i^G + F_i \cdot \left( \mathbf{m}_{pbest}^G - \mathbf{m}_i^G + \mathbf{m}_{r_1}^G - \widetilde{\mathbf{m}}_{r_2}^G \right), \tag{6}$$

where $\widetilde{\mathbf{m}}_{r_2}^G$ is a randomly selected vector from the expanded population $\mathbf{P} \cup \mathbf{A}$. The set $\mathbf{A}$ is an optional archive used to store the inferior target vectors in selection. When the size of $\mathbf{A}$ is more than the predefined number, some vectors are randomly removed.

- Adaptation of control parameters:

In JADE, the $F$ and $CR$ values are obtained from the Cauchy and Normal distributions. For the target vector $\mathbf{m}_i^G$ at $G$th iteration, its $CR$ value is calculated based on the following formula:

$$CR_i^G = randn_i \left( \mu_{CR}^G, 0.1 \right), \tag{7}$$

where $randn_i(\cdot)$ returns a Normal distribution number with mean value $\mu_{CR}^G$ and standard deviation 0.1. If $CR_i^G$ is outside of the range of $[0, 1]$, it is truncated as 0 or 1. At each iteration, the parameter $\mu_{CR}^G$ is adapted with the following equation:

$$\mu_{CR}^G = (1 - c)\mu_{CR}^{G-1} + c \, mean_A(S_{CR}), \tag{8}$$

where $mean_A(\cdot)$ returns the arithmetic mean of input data, $S_{CR}$ is a set used to store the good $CR$ values in the last iteration. $c$ is the learning rate in the 0 and 1 range. Similarly, the generation of $F$ and adaptation of $\mu_F$ are fulfilled according to the following:

$$\begin{cases} F_i^G = rand_c\left(\mu_F^G, 0.1\right) \\ \mu_F^G = (1-c)\mu_F^{G-1} + c\, c \\ mean_L(S_F) = \frac{\sum_{F \in S_F} F^2}{\sum_{F \in S_F} F} \end{cases}, \tag{9}$$

where $randc_i(\cdot)$ returns a Cauchy distribution number with location parameter $\mu_F^G$ and scale factor 0.1. $mean_L(\cdot)$ is Lehmer mean. The set $S_F$ stores the successful $F$ values, which produce better trial vectors at the last iteration.

2.1.3. The Proposed JADE

- The $CR$ generation mechanism;

As stated by Zheng et al. in [47], embedding the high-quality vectors into the mutation strategy are more likely to generate preferable directions for the search. In this work, a $CR$ generation strategy is designed to maintain the information of high-quality vectors according to the following formulas:

$$CR_i^G = \mu_{CR}^G + 0.1 \cdot \delta_i^G,$$
$$\delta_i^G = \frac{\Phi(\mathbf{m}_i^G) - mean_A(\boldsymbol{\Phi})}{mean_A(|\boldsymbol{\Phi} - mean_A(\boldsymbol{\Phi})|)}, \tag{10}$$

where $\boldsymbol{\Phi} = \left[\Phi(\mathbf{m}_1^G), \Phi(\mathbf{m}_2^G), \cdots, \Phi(\mathbf{m}_{NP}^G)\right]$ is a set of all objective values in $\mathbf{P}$. From Equation (10), for a minimization problem, a vector with fitness value less than $mean_A(\boldsymbol{\Phi})$ will obtain a $CR$ value smaller than $\mu_{CR}^G$. Assume that all vectors in $\mathbf{P}$ are sorted in an ascending manner according to fitness values, then the proposed $CR$ scheme can be illustrated as Figure 1.

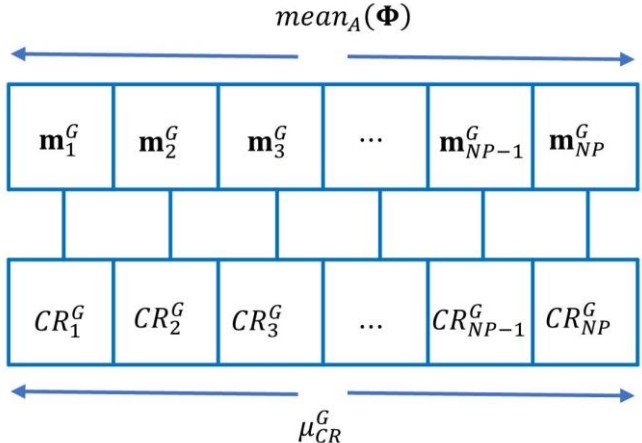

**Figure 1.** An illustration of the proposed $CR$ generation scheme.

- Rank-based mutation strategy:

In JADE, the perturbation direction is a difference vector toward $\mathbf{m}_{r_1}^G$. Zhang and Sanderson believed that the difference between $\mathbf{m}_{r_1}^G$ and $\widetilde{\mathbf{m}}_{r_2}^G$ carries the information toward the optimum [48]. Therefore, a proper choice of $\widetilde{\mathbf{m}}_{r_2}^G$ is beneficial to accelerate the convergence speed. In this work, the rank of each individual is employed to calculate the

selection probability. If the rank of the $i$th vector in the set $\mathbf{P} \cup \mathbf{A}$ is $rank\left(\overset{\sim}{\mathbf{m}}_i^G\right)$, the selection probability $p_i$ is obtained by the following formula:

$$p_i^G = \left(\frac{NP + N_A - rank\left(\overset{\sim}{\mathbf{m}}_i^G\right)}{NP + N_A}\right)^2 , \tag{11}$$

where $N_A$ represents the size of archive $\mathbf{A}$. The details of the selection for $\overset{\sim}{\mathbf{m}}_{r_2}^G$ are presented in Algorithm 1. From Equation (11) and Algorithm 1, we can see that a vector with an inferior objective value is more easily selected. Then, a possible direction toward the optimum is obtained with high probability by the difference vector $\mathbf{m}_{r_1}^G - \overset{\sim}{\mathbf{m}}_{r_2}^G$.

---

**Algorithm 1** The selection of $\overset{\sim}{\mathbf{m}}_{r2}^G$

---

Input: The index $i$ and terminal vector index $r_1$
Output: The selected vector index $r_2$
1: randomly generate an integer $r_2$ in the range of 0 and $NP + N_A$
2: while $rand(0,1) \le p_{r_2}^G$ or $r_2 == i$ or $r_2 == r_1$
3: randomly generate an integer $r_2$ in the range $[1, NP + N_A]$
4: end while

---

### 2.2. Forward Modeling of Gravity

In this subsection, the forward modeling of gravity anomalies is simply described. Traditionally, the gravity anomalies caused by the given density distribution are calculated using integration derived from Newton's law. For a simple gravity field source shown in Figure 2, the gravity data $g_z(x,z)$ at a point $(x,z)$ caused by density distribution $\Delta\rho(x,z)$ can be calculated according to the following [2]:

$$g_z(x,z) = 2\Gamma \int_S \frac{\Delta\rho(x,z)}{r^2} dxdz , \tag{12}$$

where $\Gamma$ is called a universal gravitational constant. Generally, the values of gravity anomalies are calculated using numerical integration [49,50]. Alternatively, if a proper boundary condition is given, they are also obtained by solving the Poisson equation as the following:

$$\nabla^2 u = -4\pi\Gamma \cdot \Delta\rho , \tag{13}$$

where $u$ represents the scalar gravity potential. Then, at a given point $(x,z)$, its gravity data are calculated by the following equation:

$$g_z(x,z) = -\frac{\partial u}{\partial z} , \tag{14}$$

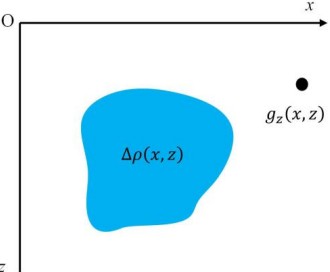

**Figure 2.** Geometry illustration of density source $\Delta\rho$ and gravity field $g_z$.

In this work, the finite volume method [51] with rectangular elements (Figure 3) is employed to solve Equation (13).

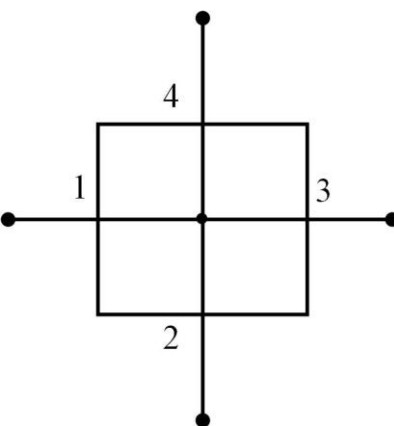

**Figure 3.** A rectangular element with face indexing.

Applying the divergence theorem to Equation (13) for all elements like Figure 3 will result in a matrix equation:

$$\mathbf{K}\mathbf{u} = \mathbf{b} ,\tag{15}$$

where $\mathbf{K}$ is a sparse matrix formed by discretization coefficients, $\mathbf{b}$ is a vector related to the density distribution, and $\mathbf{u}$ represents the unknown gravity potential values. After applying proper boundary conditions (e.g., Dirichlet boundary [52], Neumann boundary [53], and artificial boundary [54]), and modern direct solvers, the scalar gravity potential values of given density distribution are obtained. In our work, the boundary condition in [54] is employed. Then, the gravity anomaly is calculated by using the centered difference scheme [55]. If we give the forward modeling process as operator $\mathcal{G}$, and the discrete density distribution as $\mathbf{m}$ with size $M$, the forward modeling of gravity can be denoted as:

$$\mathbf{g}_z = \mathcal{G}(\mathbf{m}) ,\tag{16}$$

where $\mathbf{g}_z$ records the calculated gravity values.

*2.3. Gravity Inversion with Multiplicative Regularization*

2.3.1. Inversion Method

According to the studies shown in [10,15,56,57], the objective function for gravity inversion is comprised of data misfit and model misfit. By introducing regularization factor $\lambda$, the objective function is written as:

$$\Phi = \Phi_d + \lambda \Phi_m ,\tag{17}$$

where $\Phi$ denotes the objective function for inversion, $\Phi_d$ represents data misfit, and $\Phi_m$ represents model misfit. In the literature [41,58], the authors called Equation (17) additive regularization. In order to avoid the selection of $\lambda$, the following multiplicative regularization technique was proposed [41]:

$$\Phi = \Phi_d \cdot \Phi_m ,\tag{18}$$

Now, the Equation (18) was generalized by Aucejo and Smet to the following [45]:

$$\Phi = \Phi_d \cdot \Phi_m^{\mu} ,\tag{19}$$

where $\mu$ is an extra tuning parameter in the range $(0, +\infty)$, which is determined using a fixed-point iteration algorithm. For the gravity inversion of this paper, the following objective function using multiplicative regularization is employed:

$$\Phi(\mathbf{m}) = (\Phi_d(\mathbf{m}))^{\mu}(\Phi_m(\mathbf{m}))^{1-\mu} ,$$
$$\text{s.t. } \mathbf{m}_l \leq \mathbf{m} \leq \mathbf{m}_u , \tag{20}$$

where $\mu$ is similar to Equation (19). $\Phi_d(\mathbf{m})$ represents data misfit, which is defined as:

$$\Phi_d(\mathbf{m}) = \frac{\left\| W_d\left(\mathbf{g}_z^{obs} - \mathcal{G}(\mathbf{m})\right) \right\|_1^1}{\left\| W_d\mathbf{g}_z^{obs} \right\|_1^1} ,$$

$$W_d = diag\left( \frac{1}{|g_{z,1}^{obs}|+\epsilon}, \frac{1}{|g_{z,2}^{obs}|+\epsilon}, \cdots , \frac{1}{|g_{z,N}^{obs}|+\epsilon} \right), \tag{21}$$

$$\epsilon = std\left( \left| \mathbf{g}_z^{obs} \right| \right) ,$$

where $\mathbf{g}_z^{obs}$ represents the observation data vector with size $N$. In $\mathbf{g}_z^{obs}$, the $i$th entry is $g_{z,i}^{obs}$, $\epsilon$ is used to avoid the dominator being zero. It is noted that $\Phi_d(\mathbf{m})$ is a weighted L1-norm formed by observed data $\mathbf{g}_z^{obs}$ and predicted data $\mathcal{G}(\mathbf{m})$. The reason for employing L1-norm in data misfit is that can suppress the outlier well in observation data [59]. In addition, In Equation (20), the model misfit $\Phi_m(\mathbf{m})$ is calculated using the following L1-norm distance:

$$\Phi_m(\mathbf{m}) = \sum_{i=1}^{M} W_{m,i}|m_i - m_{\text{ref},i}| , \tag{22}$$

where $\mathbf{m}_{\text{ref}} = (m_{\text{ref},1}, m_{\text{ref},2}, \cdots, m_{\text{ref},M})$ denotes the reference model, $W_{m,i}$ is the weight parameter of $i$th entry in vector $\mathbf{m}$. As we know, the L1-norm model misfit is a good approximation of L0 [60], which is effective to retain the sparseness of inverted models. The $\mathbf{W}_m = (W_{m,1}, W_{m,2}, \cdots, W_{m,M})$ is obtained according to the following formula [57]:

$$W_{m,i} = \frac{W_{z,i} V_i}{\sum_{k=1}^{M} W_{z,k} V_k} ,$$
$$W_{z,i} = \frac{1}{D_{z,i}^{\alpha}} , \tag{23}$$

where $V_i$ represents the volume size of element $i$, $W_{z,i}$ a parameter related to the depth of $i$th element, and $D_{z,i}$ denotes the depth of element $i$. $\alpha$ is a decreasing factor and set to 1 for the gravity inversion. Finally, according to the equations from (21) to (23), the objective function can be rewritten as the following:

$$\Phi(\mathbf{m}) = \left( \frac{\left\| W_d\left(\mathbf{g}_z^{obs} - \mathcal{G}(\mathbf{m})\right) \right\|_1^1}{\left\| W_d\mathbf{g}_z^{obs} \right\|_1^1} \right)^{\mu} \left( \sum_{i=1}^{M} W_{m,i}|m_i - m_{\text{ref},i}| \right)^{1-\mu} , \tag{24}$$

Observations from Equations (19) and (24) show that the extra weight parameter is designed to give a proper weight for the model misfit function. In Equation (24), if $\mu \to 0$, the $\Phi(\mathbf{m})$ is dominated by the model misfit. On the contrary, if $\mu \to 1$, the data misfit will dominate the value of $\Phi(\mathbf{m})$. Compared with additive regularization, although both regularization factor and weight parameter $\mu$ have an influence on the balance of model misfit and data misfit, the parameter $\mu$ in (24) is independent of the values of data and model misfit. So, it can be more easily adapted in the process of searching.

### 2.3.2. Adaptation of Extra Weight Parameter $\mu$

Although swarm intelligence algorithms like DE and PSO are adept in global search, an inappropriate selection of extra weight parameter $\mu$ or regularization factor still may result in unreasonable solutions. In addition, according to the study in [61], the effect of

model misfit should be added when the data misfit is less than a predefined tolerance. Here, we design an adaptation scheme for parameter $\mu$ according to the following formulas:

$$\mu^G = \begin{cases} \min\left(1, 1.5 \cdot \mu^{G-1}\right), \; if \; q^G \geq 1 \\ \max\left(0.95, q^G\right) \cdot \mu^{G-1}, \; \text{otherwise} \end{cases}$$

$$q^G = \left(\frac{\Phi_{d,\text{mean}}^{G-1}}{\Phi_{d,\text{mean}}^{G-2}}\right)^2, \tag{25}$$

$$\Phi_{d,\text{mean}}^k = \frac{\sum_{i=1}^{NP} \Phi_d\left(\mathbf{m}_i^k\right)}{NP}, k = G-1, G-2,$$

where $\Phi_{d,\text{mean}}$ denotes the mean value of the obtained data misfit values in the DE population, and $q_G$ is the ratio of $\Phi_{d,\text{mean}}^{G-1}$ and $\Phi_{d,\text{mean}}^{G-2}$. According to Equation (25), if $q_G$ is more than or equal to 1 it implies that the current parameter $\mu$ may fail to decrease the data misfit. In this situation, decreasing the value of $\mu$ will accelerate the convergence speed. In addition, when the mean of data misfit $\Phi_{d,\text{mean}}$ stably decease, increasing the parameter $\mu$ will make the optimization algorithm prone to optimize the prior information in model misfit. In this case, the $q_G$ is less than 1, which makes the $\mu$ in Equation (25) increase. Moreover, the value of $\mu$ is initialized as 0.5 to ensure that the optimization algorithm simultaneously optimizes the data and model misfits at the beginning of the search. In summary, from the above discussion, the proposed adaptation of $\mu$ will balance the effect of data and model misfits in the whole search process.

### 2.3.3. Smooth Strategy for Gravity Inversion

To obtain a smooth density distribution, it is better to smooth its perturbation direction when using DE for inverting the physical property of a field source [18], such as:

$$\mathbf{v}_i^G = \mathbf{m}_i^G + F_i \cdot \left(\mathbf{m}_{pbest}^G - \mathbf{m}_i^G + \mathbf{S}^4\left(\mathbf{m}_{r_1}^G - \tilde{\mathbf{m}}_{r_2}^G\right)\right), \tag{26}$$

where $\mathbf{S}$ is a sparse matrix comprised of the weight parameters of the moving average (see Figure 4).

| | | |
|:---:|:---:|:---:|
| $\dfrac{1}{16}$ | $\dfrac{2}{16}$ | $\dfrac{1}{16}$ |
| $\dfrac{2}{16}$ | $\dfrac{4}{16}$ | $\dfrac{2}{16}$ |
| $\dfrac{1}{16}$ | $\dfrac{2}{16}$ | $\dfrac{1}{16}$ |

**Figure 4.** An illustration for the weight parameters in sparse matrix $\mathbf{S}$.

### 2.3.4. The Framework of Gravity Inversion

In this subsection, the framework of gravity inversion with multiplicative regularization will be presented in detail, and its workflow is shown in Figure 5.

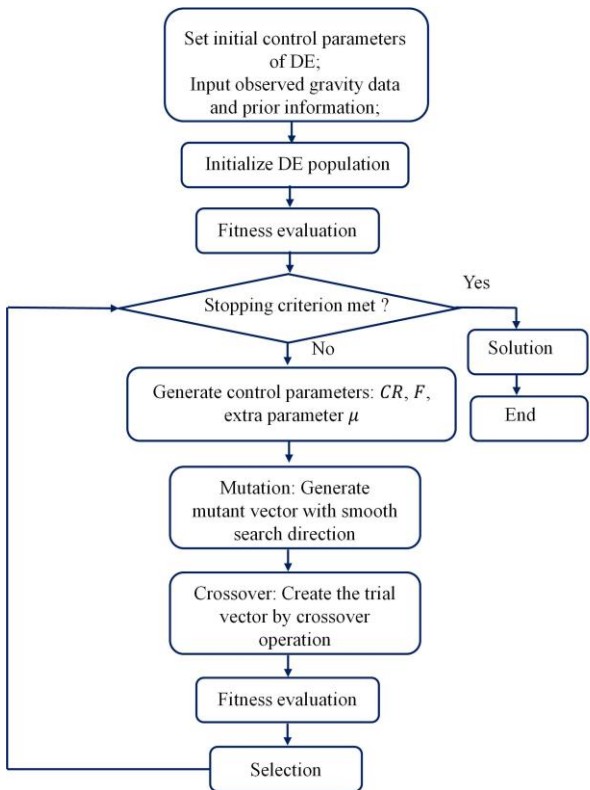

**Figure 5.** Workflow of gravity inversion with multiplicative regularization using DE.

Preparatory work: In this step, the preprocessing of gravity data should be performed including data denoising, separation of local and global fields, etc. In particular, to decrease the multiplicity of gravity inversion, a reference model is significantly useful to reduce the non-uniqueness of inversion results if it can be given. The extra weight parameter $\mu = 0.5$. Moreover, the initial parameters for DE are $\mu_{CR} = 0.5$, $\mu_F = 0.5$, and $p\text{best} = ceil(0.05 \cdot NP)$.

1.  Population initialization: if have no reference model, the initial solutions are generated with the following formula:

$$m_{i,j}^0 = 0.001 \cdot rand(0,1)\,, \tag{27}$$

   Otherwise, the following equation is applied:

$$m_{i,j}^0 = m_{ref,j} + 0.001 \cdot rand(0,1). \tag{28}$$

2.  Fitness evaluation: calculate the initial values of data misfit and model misfit, then obtain the objective value of each model according to Equation (20);
3.  Stop criterion check: if the termination condition is met, output all models and associated predicted gravity data;
4.  Generation of control parameters: the $CR$ values are obtained according to the formula (10). For scale factor $F$, it is sampled from $(0, 1]$ according to the Cauchy distribution shown in Equation (9). Moreover, the extra weight factor is calculated by using Equation (25);
5.  Mutation: for the presented work, the subindex of $\mathbf{m}_{r_2}^G$ is selected by Algorithm 1, then the mutant vector is generated by Equation (26);
6.  Crossover: in this step, the trial vector of target vector $\mathbf{m}_i^G$ is formed through applying the way of Equation (3);
7.  Fitness evaluation: for each trial vector, its objective value is estimated by Equation (20);

8.  Selection: the comparison is conducted between a trial vector and its target vector according to Equation (4). The better one is saved into the new population. In addition, the values of $\mu_{CR}$ and $\mu_F$ are updated with Equation (8) and Equation (9), respectively;
9.  Return to 4: if the termination condition is not met, repeat steps 5 to 9.

## 3. Test and Application

In this section, the proposed method in the above section is tested and verified by synthetic and field data. For the synthetic test, four models (shown in Figure 6) with a density of 1 g/cm$^3$ are designed. The noise-free gravity data are displayed in Figure 7. The space distance and number of data are 5 m and 81, respectively. However, when inverting the gravity data, the point space is set to 10 m, and the element size in the $z$ direction proportionally varies with the depth value.

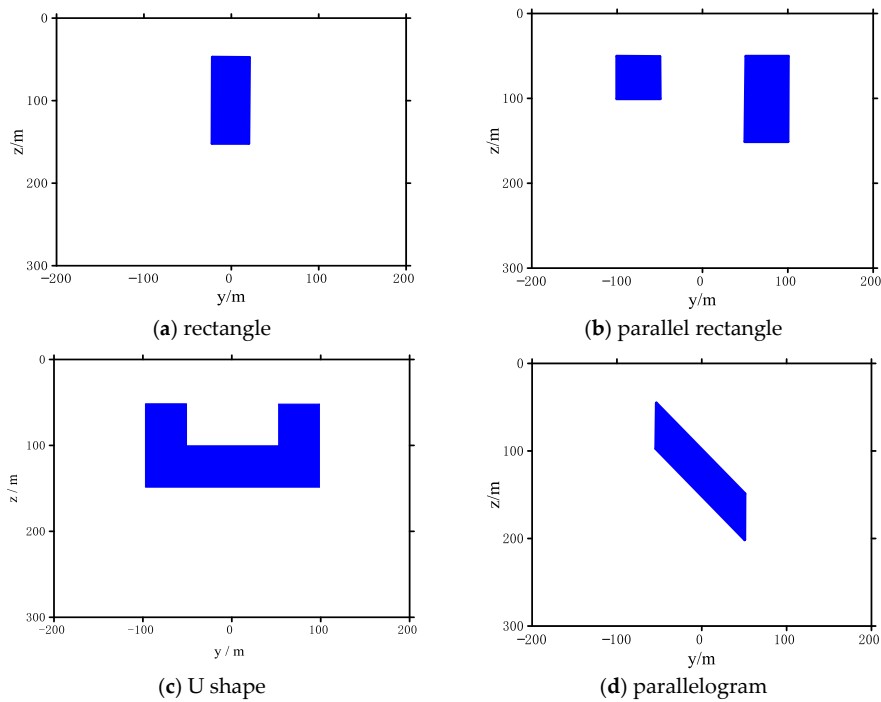

(**a**) rectangle

(**b**) parallel rectangle

(**c**) U shape

(**d**) parallelogram

**Figure 6.** Synthetic models designed for gravity inversion. The blue shapes represent the density distribution with value 1 g/cm$^3$.

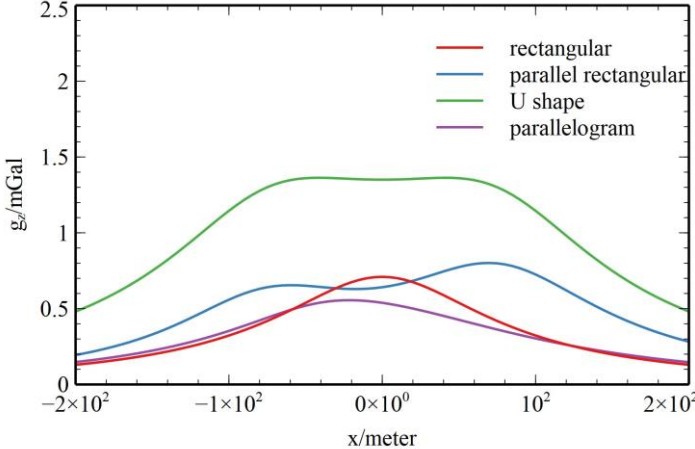

**Figure 7.** The curves of gravity data for designed models.

*3.1. The Effectiveness of Proposed Modifications for JADE*

For a fair comparison, the parameter $NP$ in DE is set to 100, the termination condition is maximum iterations (300). In addition, the ranges for the search are 0 g/cm$^3$ for the lower boundary and 1.1 g/cm$^3$ for the upper boundary. According to the discussion in previous sections, there are two modifications for improving JADE such as generation of the $CR$ value and selection of the index $\widetilde{\mathbf{m}}_{r_2}$. In this subsection, the proposed modifications are investigated and analyzed. Therefore, several variants of the proposed algorithm are used:

Version 1: the JADE algorithm proposed by Zhang and Sanderson. In this version, the proposed modifications are not applied. It is denoted by IADE-1.

Version 2: based on Version 1, the selection method in Algorithm 1 for the subindex of $\widetilde{\mathbf{m}}_{r2}$ is used. This version is represented by IADE-2.

Version 3: based on Version 2, the generation scheme of $CR$ proposed in Section 2.1.3 is added. This version is represented by IADE.

Considering that the forward modeling is time-consuming, each algorithm is independently run 10 times using noise-free data, and the experimental results are reported in Table 1 according to the obtained mean (Mean) and standard deviation (Std) from data misfit values. The best results are illustrated with bold font. From Table 1, it is concluded that the IADE algorithm with all modifications obtains the best results on four models, which implies that IADE is superior to its variants. Moreover, IADE-2 defeats IADE-1 in three cases, which means that the modified mutation strategy is beneficial to enhance the solution quality. In Figure 8, the convergence curves of IADE, IADE-1, and IADE-2 are displayed. It can be seen from Figure 8 that IADE converges much faster than IADE-1 and IADE-2, and obtains inversion solutions with higher accuracy. To draw a statistical conclusion, the significant difference between IADE, IADE-1, and IADE-2 are evaluated using the multi-problem Wilcoxon signed-rank test. The related results are reported in Table 2. According to Table 2, it is seen that IADE is statistically better than IADE and IADE-2 at a significance level $\alpha = 0.05$. Simultaneously, IADE-1 performs better than IADE-1 since it acquires a higher $R^+$ value than $R^-$. In [23], the authors used the standard deviation of the inverted models to measure the uncertainty of inversion. In our view, the standard deviation from inverted models is related to the mean model of all solutions. Therefore, the uncertainty can be represented if we output the mean model as the final solution. From this point, the mean model of the obtained solutions is listed in Figures A1–A3 in Appendix A. From these graphs, we can conclude that that the IADE obtains superior solutions than IADE-1 and IADE-2 according to the difference between the true density distribution (drawn by a black polygon) and the inverted models.

**Table 1.** Experimental results of IADE-1, IADE-2, and IADE based on the obtained mean and standard deviation calculated from data misfit on different models.

| Models | IADE-1 (Mean $\pm$ Std) | IADE-2 (Mean $\pm$ Std) | IADE (Mean $\pm$ Std) |
|---|---|---|---|
| rectangular | $5.01 \times 10^{-3} \pm 5.63 \times 10^{-3}$ | $7.01 \times 10^{-3} \pm 4.36 \times 10^{-3}$ | $\mathbf{2.78 \times 10^{-3} \pm 1.50 \times 10^{-3}}$ |
| parallel rectangular | $5.40 \times 10^{-2} \pm 2.52 \times 10^{-2}$ | $4.35 \times 10^{-2} \pm 1.56 \times 10^{-2}$ | $\mathbf{4.75 \times 10^{-3} \pm 1.58 \times 10^{-3}}$ |
| U shape | $3.10 \times 10^{-2} \pm 1.64 \times 10^{-2}$ | $2.22 \times 10^{-2} \pm 1.37 \times 10^{-2}$ | $\mathbf{1.84 \times 10^{-3} \pm 1.05 \times 10^{-3}}$ |
| parallelogram | $2.24 \times 10^{-2} \pm 1.05 \times 10^{-2}$ | $1.24 \times 10^{-2} \pm 8.64 \times 10^{-3}$ | $\mathbf{4.95 \times 10^{-3} \pm 9.32 \times 10^{-3}}$ |

In summary, based on the results in Tables 1 and 2, and solutions presented in Figures A1–A3 in Appendix A, we can conclude that the new algorithm with proposed modifications is more powerful and stable than its variants. Therefore, according to the solution quality and convergence speed, the proposed modifications are useful for improving the optimization performance of DE, and the adaptive DE algorithm with these improvements is superior and robust in comparison with its variants.

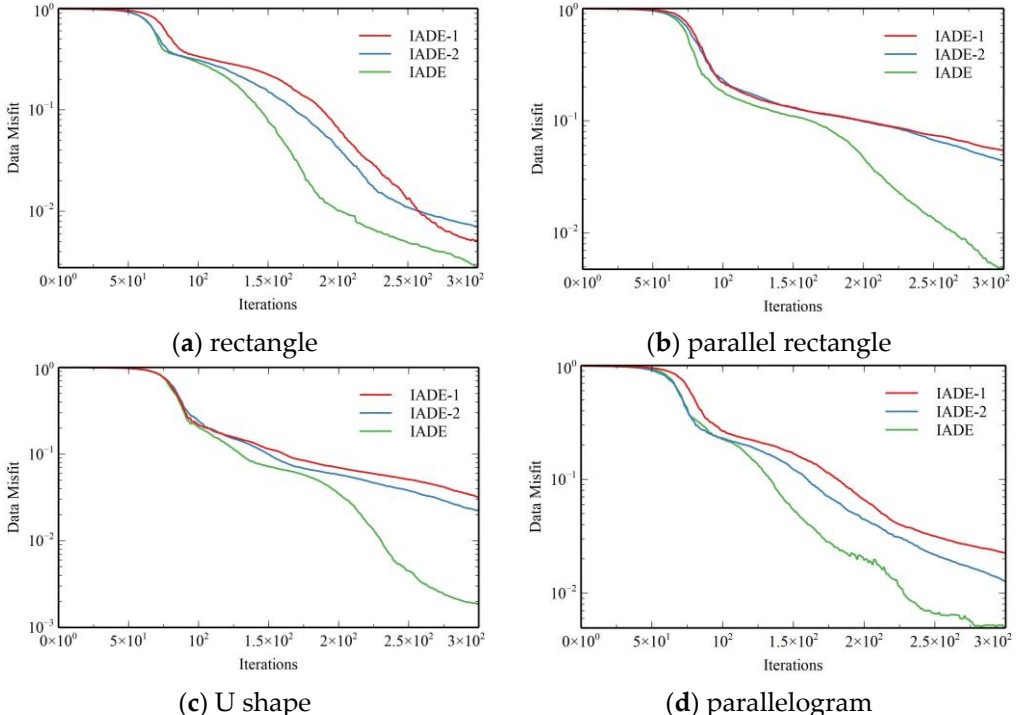

(**a**) rectangle                          (**b**) parallel rectangle

(**c**) U shape                            (**d**) parallelogram

**Figure 8.** Convergence curves of data misfit for different algorithms on different models.

**Table 2.** Results of multiple-problem Wilcoxon's test of IADE, IADE-1, and IADE-2 on designed models.

| Algorithm | $R^+$ | $R^-$ | *p*-Value | $\alpha = 0.05$ |
|---|---|---|---|---|
| IADE vs. IADE-1 | 10 | 0 | 0.04461 | yes |
| IADE vs. IADE-2 | 10 | 0 | 0.04461 | yes |
| IADE-2 vs. IADE-1 | 5 | 1 | 0.181449 | no |

### 3.2. Parameter Setting Study of μ

In Section 2.3.2, an extra parameter $\mu$ is introduced to balance the effect of data and model misfit. The convergence curves of $\mu$ for IADE on different models are displayed in Figure 9. In terms of Equation (24), the objective for inversion is dominated by the data misfit function when the value of $\mu$ is more than 0.5. Then, Equation (25) reveals that the parameter $\mu$ will increase with the increment of the data misfit. At the beginning of the search, the values of data misfit in the DE population are large since the initial models are far away from the true solution. In this case, optimizing the model misfit is not useful for minimizing the errors between the observed data and the predicted. From Figure 9, one can conclude that, in order to close the gap between the observed and the predicted, the value of $\mu$ for all designed models rapidly increases at the early stage of the search. After which, the $\mu$ converges to small values close to zero so as to make the obtained solutions fit the prior information.

### 3.3. The Noise Effect in Gravity Data

Considering that the gravity data in practical examples are contaminated by noise from the instrument and environment, it is therefore necessary to evaluate the influence caused by noise on inverted models. In this work, the noise is added according to the following formula,

$$g_{z,i}^{obs} = g_{z,i} + \sigma \cdot std(|\mathbf{g}_z|) \cdot randn(0,1), \tag{29}$$

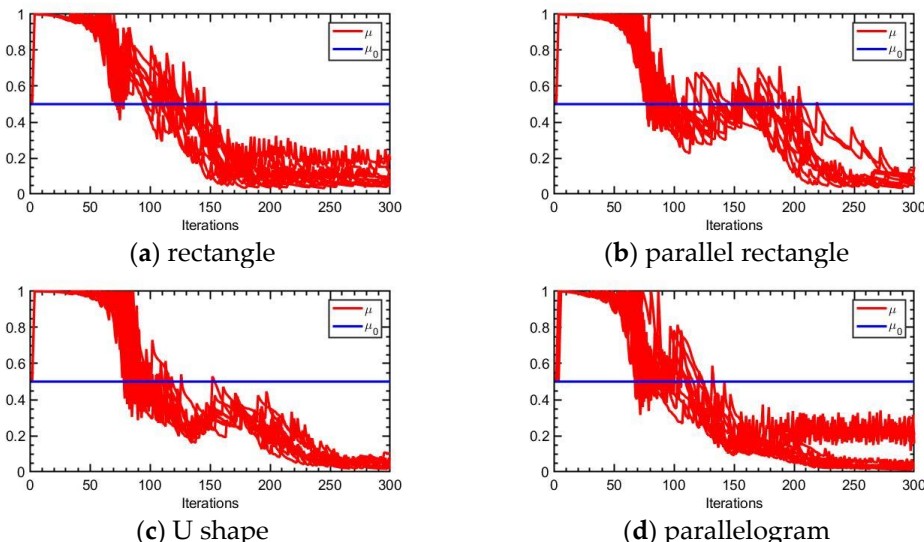

**Figure 9.** Convergence curves of μ for IADE on different models with 10 independent runs.

where $\sigma$ represents the noise level (i.e., 1%, 5%, and 10%), $g_i$ is noise-free gravity data, and $randn(\cdot)$ means a random number with Normal distribution. For each type of gravity data with noise, 10 independent runs with a maximum iteration of 300 are conducted. In Figure 10, the gravity data calculated from the U shape with/without noise are presented. In Table 3, the mean data misfit and the associated standard deviation according to the obtained best models at each run are reported. We can see from Table 3 that data misfit values increase with the increment of noise level, which implies that the overfit of observation data is avoided when running the gravity inversion with the proposed optimization algorithm and extra parameter adaptation. The inverted results with different noise levels are listed in Figure A4 in Appendix A. Compared with Figure A1c, the inverted models with noise data are in accord with the noise-free case, which indicates that the proposed inversion scheme with multiplicative regularization is robust and stable.

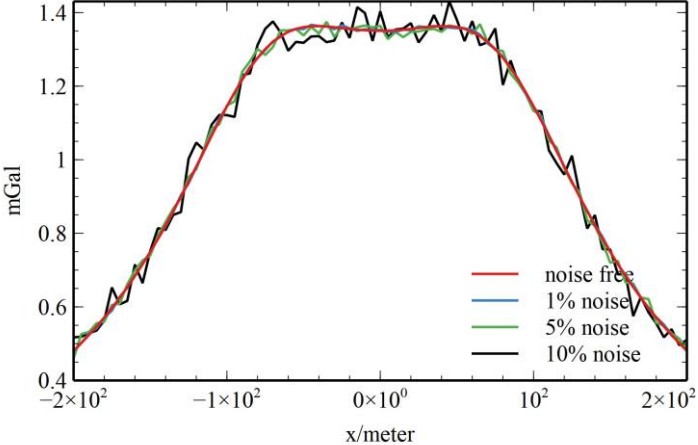

**Figure 10.** The gravity data with different noise levels for the U-shape model.

**Table 3.** The mean and standard deviation of data misfit on different noise levels.

| Model | 1% Noise (Mean ± Std) | 5% Noise (Mean ± Std) | 10% Noise (Mean ± Std) |
|---|---|---|---|
| U shape | $2.59 \times 10^{-3} \pm 4.43 \times 10^{-4}$ | $1.06 \times 10^{-2} \pm 3.50 \times 10^{-4}$ | $2.27 \times 10^{-2} \pm 2.68 \times 10^{-3}$ |

*3.4. Field Example 1: Prospecting Ultrabasic Rocks in the Poshi Cu–Ni Deposit, Xinjiang, China*

In the Xinjiang region, Pobei rocks are sited at the Beishan rift belt in the northeast of the Tarim platform, and is one of the most significant zones for exploring Cu-Ni metallogeny (Figure 11). When searching for magmatic copper–nickel deposits, a significant premise is discovering the basic–ultrabasic rocks of Pobei, which also are its main metallogenic geology elements. So, it is significantly crucial to evaluate the spatial distribution so as to prospect the Cu-Ni deposits in Pobei [62,63]. From Figure 11, Poshi basic–ultrabasic rocks are a part of the Pobei rocks, which mainly are located at the center of the basic–ultrabasic rocks. In addition, the ultrabasic rocks are distributed with an asymmetrical ellipse and strike NE–SW.

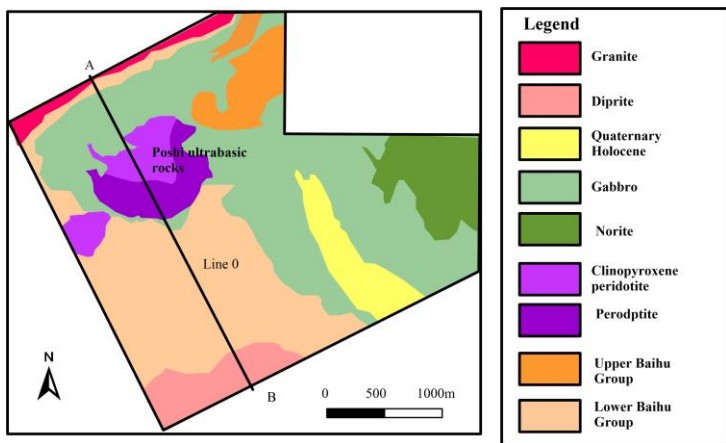

**Figure 11.** The location of Line 0 and the modified geological map of the Poshi deposit. Modified from [64,65].

The density values measured from rock and drill core samples are listed by Liu et al. in [65]. From their analysis, the largest density (i.e., >3.0 g/cm$^3$) is achieved by olivine gabbro. The peridotite has a similar density value to marble. The smallest density values (i.e., <2.7 g/cm$^3$) are found in biotite quartz schist, acid dioritic porphyrite, and biotite potash feldspar granite. In addition, the density of ultrabasic rocks (i.e., 2.6–2.8 g/cm$^3$) is lower in comparison with that of ore-bearing basic–ultrabasic rocks, gabbro, serpentine, and garnet skarn (i.e., 2.9–3.1 g/cm$^3$). Consequently, the rock formation comprised of gabbro will cause a positive anomaly. On the contrary, the contact region between different rocks and ultrabasic rocks including peridotite and serpentinized pyroxene peridotite will dominate the negative anomalies in gravity data.

In order to investigate the spatial distribution of ultrabasic rocks, Line 0 with a profile length 2400 m is designed across the ultrabasic rocks as shown in Figure 11. The number of gravity data along Line 0 is 60 as the data are sampled every 40 m. As displayed by Figures 11 and 12a, the observed data at the area where the ultrabasic rocks are exposed are lower than in other regions. Similar to [64,65], the prospecting depth is set to 1200 m. After running the presented inversion algorithm, the calculated gravity data fit the observed well (Figure 12a), and the inverted density distribution is presented in Figure 12b. Compared with other previous studies ([64,65]), a good correlation can be found between ours and these methods.

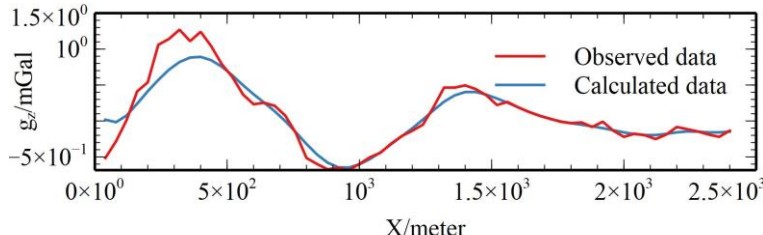

(**a**) A comparison between the observed data and calculated data.

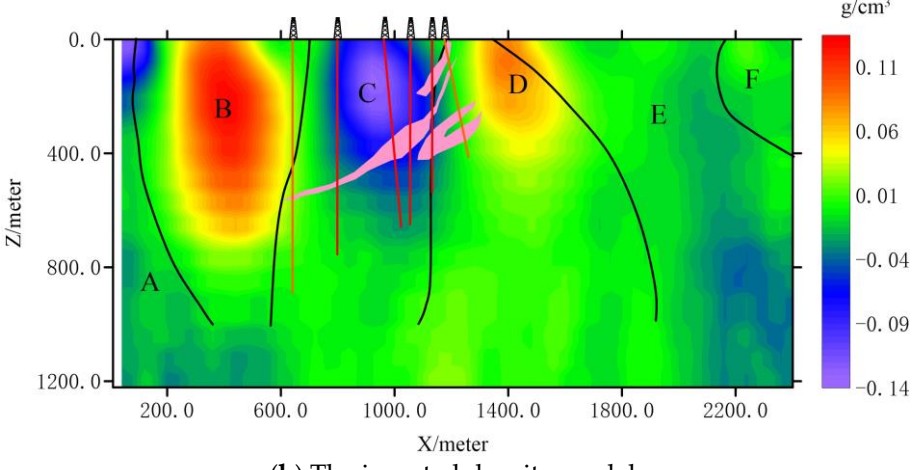

(**b**) The inverted density model.

**Figure 12.** The inversion results of Line L0 with geological section and drilling wells [65], where A, B, C, D, E, and F represent granite, earlier gabbro, latter gabbro, schist, and diorite rock units, respectively. Pink blocks denote Nickel ore bodies.

According to the inverted results, the obtained model makes a good approximation to the information indicated by geology and drilling wells. Furthermore, from Figure 12b, it is clear that the subsurface cross section is comprised of six regions with different lithology units based on the geology map and inverted density distribution. More specifically, both zone B and zone D possess high residual density values, but zone B is slightly higher than that of D. Combining Figures 11 and 12b, the ultrabasic peridotite with lower residual gravity anomaly is depicted as having lower density values by zone C. This means that the lower residual gravity and lower density are the features of ultrabasic peridotite. For zone A, which is located at the most NW area in Figure 11, its residual density value is slightly lower than zones E and F. According to the mean density values of rock samples listed by Liu et al. [65], the density of samples from granite in zone A has no noticeable difference with the biotite plagioclase quartz schist at zone E and diorite in F, but it is significantly different with zone B. Moreover, the gravity data at the proximity of A are lower than at zone B. This implies that there may exist a source distribution with a density value lower than the one of zone B. Thus, a relatively reasonable interpretation is that the negative gravity anomalies of zone A are caused by the underground granite, although there exists an over and underestimation near the border. For zones E and F, the intrusions of diorite at F are slightly higher than the metamorphic rocks at E, although their difference is unclear. In a word, according to the inversion results shown in Figure 12, one can conclude that there exist six different rock units along Line 0, which are A—granite, B—earlier gabbro, C—peridotite, D—latter gabbro, E—metamorphic rocks, and F—diorite. Moreover, the density values of A and C are low, B and D are high, and a moderate density value is obtained by E and F [65].

*3.5. Field Example 2: Iron Deposit Prospection of Shihe, Shanxi, China*

The Shihe iron deposit is situated in Shanxi, China. The outcropped rocks in this mining area are entirely covered by Quaternary sediments of the Malan group, ranging

in depth from 240 to 340 m (Figure 13). Geologically, the Shihe deposit is controlled by the uplift of the Hengshan–Wutaishan in the Lvliang–Taihang fault block. The density values of rock and drill core samples are presented in Table 4. From Table 4, the Quaternary sediment samples exhibit the lowest density value (i.e., 1.42–1.58 g/cm$^3$), while Plagioclase amphibolite, Hornblende–Plagioclase, Biotite leptynite, and Jinggang group have similar moderate density values. The highest density value is found in magnetite quartzite (i.e., >3.13 g/cm$^3$), which is the iron mineral in our study area. Therefore, by combining Table 4 and the observed data in Figure 14a, it is likely that the positive residual anomalies are caused by magnetite quartzite.

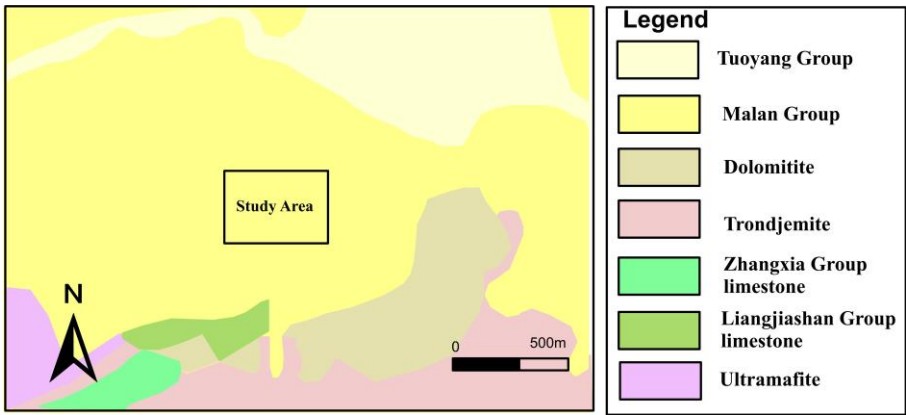

**Figure 13.** The simple geological map of Shihe deposit in Shanxi, China. Modified according to the geology map provided by Shanxi Institute of Geophysical and Geochemical Exploration, China.

**Table 4.** Density measurements of rock samples collected from outcrop and boreholes in Shihe deposit. Measured data are provided by Shanxi Institute of Geophysical and Geochemical Exploration, China.

| Rocks | Sample Number | Rang $\rho$ g/cm$^3$ | Mean $\rho$ g/cm$^3$ |
|---|---|---|---|
| Magnetite quartzite | 30 | 3.13–3.46 | 3.16 |
| Plagioclase amphibolite | 41 | 2.48–2.94 | 2.78 |
| Hornblende-Plagioclase | 7 | 2.43–2.81 | 2.65 |
| Biotite leptynite | 28 | 2.57–2.90 | 2.88 |
| Quaternary sediments | 6 | 1.42–1.58 | 1.53 |
| Jinggang group | 106 | 2.43–3.46 | 2.78 |

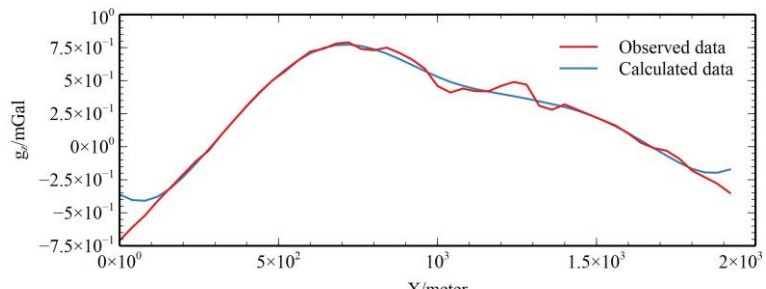

(**a**) A comparison between the observed data and calculated data.

**Figure 14.** *Cont.*

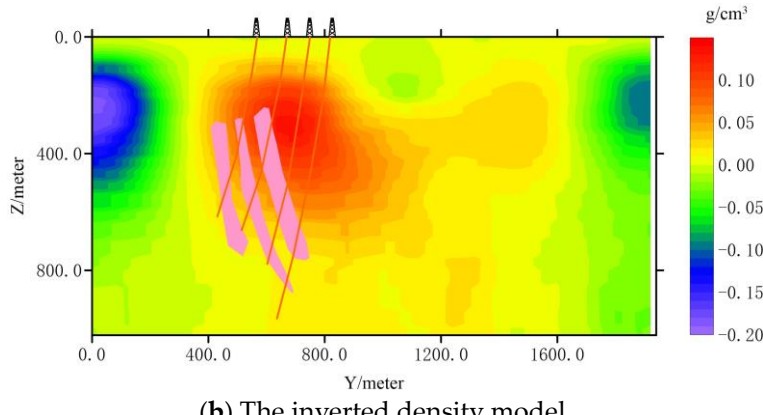

(**b**) The inverted density model.

**Figure 14.** The inversion results of gravity data with drilling wells in Shihe. Pink blocks denote iron ore bodies.

In order to investigate the spatial distribution of magnetite quartzite, the ground measurements of gravity data were acquired by the Shanxi Institute of Geophysical and Geochemical Exploration. A gravity profile with a length of 1870 m and an azimuth angle of 165° was designed across the study area, as shown in Figure 13. During the gravity inversion process, the maximum prospecting depth was set as 0.5 times the profile length. After running the proposed inversion algorithm, the calculated gravity data and the inverted density distribution are displayed in Figure 14. It is clear that a good correlation exists between the inverted model and the borehole in formation.

## 4. Conclusions

In this work, the main focuses are introducing multiplicative regularization and improving adaptive differential evolution for the inversion of gravity data. Firstly, to accelerate the convergence speed of DE, a new mutation strategy with inferior vector selection in terms of the rank values of all vectors is proposed. In addition, the search abilities of local and global are balanced by the designed crossover rate mechanism, which calculates the *CR* value without relying on the probability distribution. Finally, the multiplicative with an extra weight parameter is introduced since it does not require the regularization parameter in the search process.

For our developed adaptive DE, the results of synthetic models indicate that, according to the solution quality, convergence speed, and robustness, it is superior to the original version. Additionally, by using an extra weight parameter, the optimization process for the inversion objective is adapted by the change in data misfit. From the inverted results, the proposed adaptation method is effective for solving gravity inversion. Furthermore, the proposed algorithm is evaluated by two practical examples in China, the obtained models show a high correlation with the known information, which indicates that the multiplicative regularization inversion with adaptive differential evolution is stable and robust.

Our future work will extend the proposed optimization algorithm and multiplicative regularization for solving the inverse problems of other geophysical methods.

**Author Contributions:** Conceptualization, L.C.; methodology, L.C.; software, L.C.; Formal analysis, T.X.; Resources, Y.L.; writing—original draft preparation, L.C.; writing—review and editing, A.W.M. and X.H.; visualization, W.D.; funding acquisition, W.D. All authors have read and agreed to the published version of the manuscript.

**Funding:** This research was funded by the National Natural Science Foundation of China with grant number 41904129.

**Data Availability Statement:** The data of this study are available from the authors upon request.

**Acknowledgments:** The authors wish to acknowledge the National Natural Science Foundation of China (Grant no. 41904129). Moreover, thanks are due to Shanxi Institute of Geophysical and Geochemical Exploration for providing gravity data of the Shihe deposit.

**Conflicts of Interest:** The authors declare no conflict of interest.

## Appendix A　Inversion Results of IADE and Its Variants

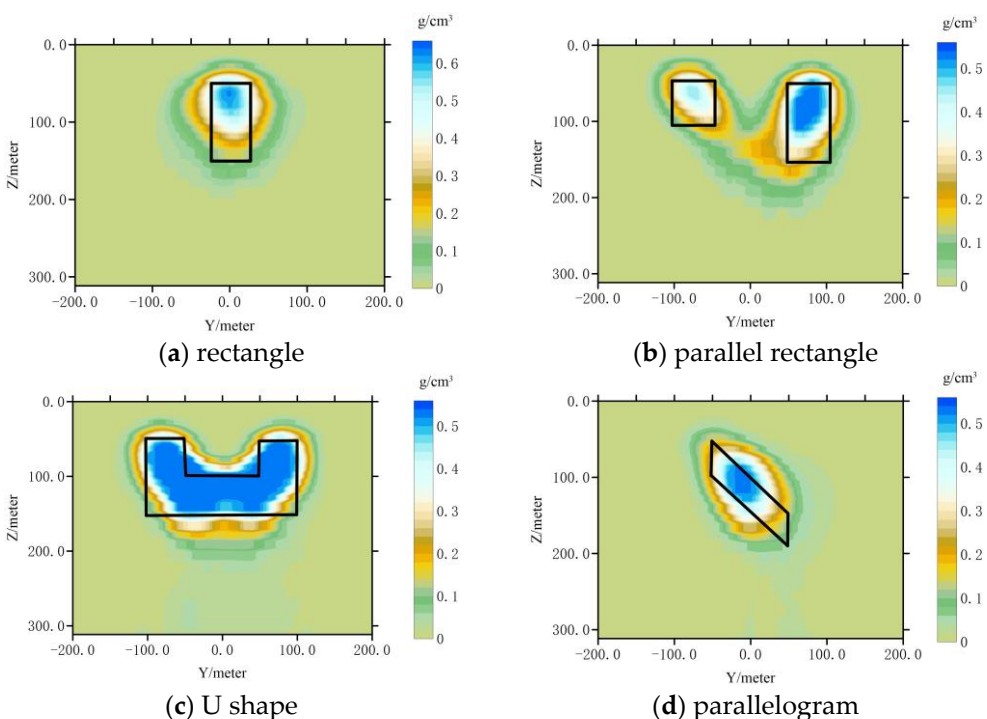

(**a**) rectangle　　　　　　　　　　　　(**b**) parallel rectangle

(**c**) U shape　　　　　　　　　　　　　(**d**) parallelogram

**Figure A1.** Gravity inversion with multiplicative regularization on designed models by using IADE. Black squares denote the designed models.

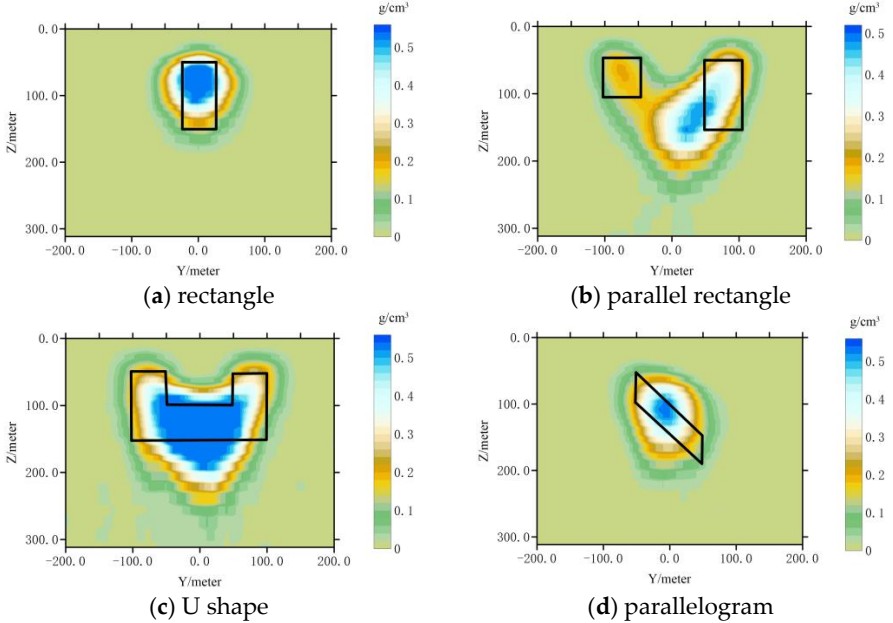

(**a**) rectangle　　　　　　　　　　　　(**b**) parallel rectangle

(**c**) U shape　　　　　　　　　　　　　(**d**) parallelogram

**Figure A2.** Gravity inversion with multiplicative regularization on designed models by using IADE-1. Black squares denote the designed models.

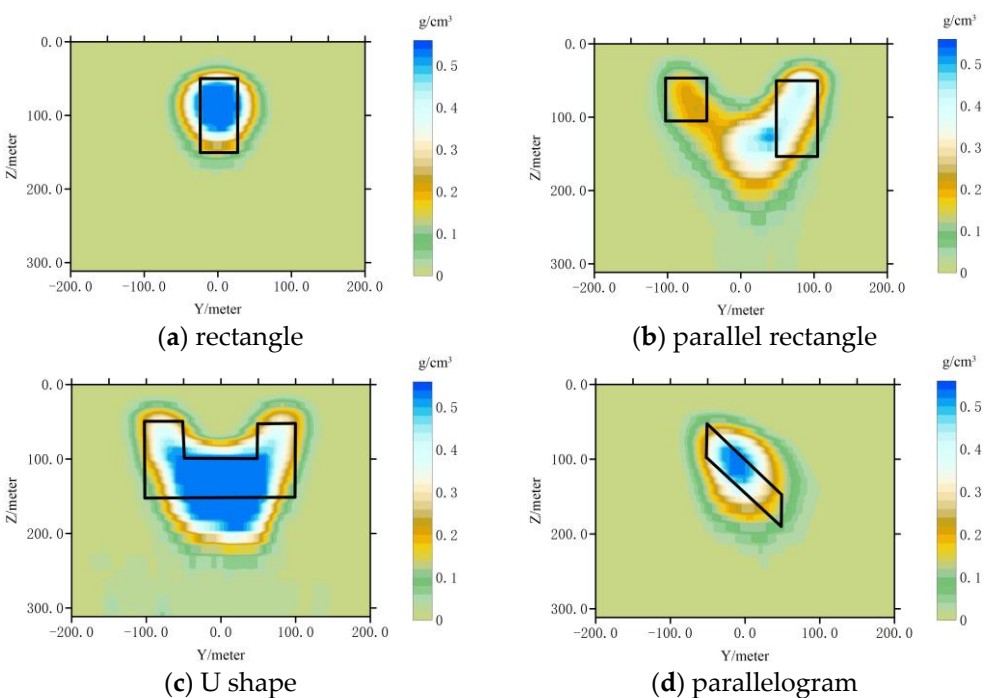

**Figure A3.** Gravity inversion with multiplicative regularization on designed models by using IADE-2. Black squares denote the designed models.

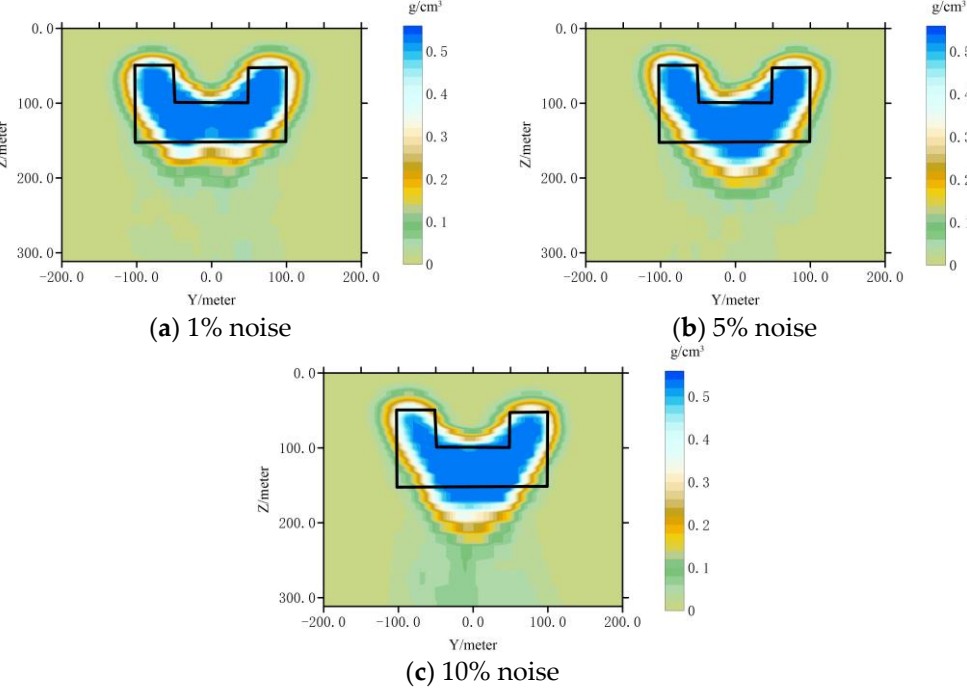

**Figure A4.** The inverted models with different noise levels for the U-shape model. Black squares denote the designed U shape model.

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
