# Peer review of "Inversion of Gravity Data with Multiplicative Regularization Using an Improved Adaptive Differential Evolution"

_minerals, doi:10.3390/min13081027_

Round 1
Reviewer 1 Report
The work presented for gravity anomaly interpretation using DE. However, I have seen that the authors have missed many important references where Optimization algorithms such as PSO, VFSA, ACO, and GP. etc which have been applied in the gravity anomaly interpretation are missing in the literature from the recent works.
The authors are encouraged to see some relevant work in interpreting subsurface idealized structures using various global optimization algorithms and make a small review in the Introduction section and how the present work is much better in terms of interpretation.
Quality is good but it needs to be polished.
Reviewer 2 Report
Please refer to the attached document.

The quality of the English language is poor. Verb tenses are often misused, and terms are employed that, although conveying an approximate meaning, are not suitable. I highly recommend seeking the assistance of a native English speaker to review the entire text thoroughly.
Round 2
Reviewer 2 Report
Please refer to the comments in the attached pdf
